# Concurrent Determination of Tigecycline, Tetracyclines and Their 4-Epimer Derivatives in Chicken Muscle Isolated from a Reversed-Phase Chromatography System Using Tandem Mass Spectrometry

**DOI:** 10.3390/molecules27196139

**Published:** 2022-09-20

**Authors:** Yawen Guo, Zhaoyuan He, Pengfei Gao, Shuyu Liu, Yali Zhu, Kaizhou Xie, Yuhao Dong

**Affiliations:** 1College of Animal Science and Technology, Yangzhou University, Yangzhou 225009, China; 2Joint International Research Laboratory of Agriculture & Agri-Product Safety, Yangzhou University, Yangzhou 225009, China; 3College of Veterinary Medicine, Nanjing Agricultural University, Nanjing 210095, China

**Keywords:** tigecycline, tetracyclines, residues, reversed-phase column, HPLC–MS/MS

## Abstract

A quantitative and qualitative method using a high-performance liquid chromatography–tandem mass spectrometry (HPLC–MS/MS) detection approach was developed and validated for the analysis of tigecycline, four tetracyclines and their three 4-epimer derivatives in chicken muscle. Samples were extracted repeatedly with 0.1 mol/L Na_2_EDTA–McIlvaine buffer solution. After vortexing, centrifugation, solid-phase extraction, evaporation and reconstitution, the aliquots were separated using a C8 reversed-phase column (50 mm × 2.1 mm, 5 µm) with a binary solvent system consisting of methanol and 0.01 mol/L trichloroacetic acid aqueous solution. The typical validation parameters were evaluated in accordance with the acceptance criteria detailed in the guidelines of the EU Commission Decision 2002/657/EC and the U.S. Food and Drug Administration Bioanalytical Method Validation 05/24/18. The matrix-matched calibration curve was linear over the concentration range from the limit of quantitation (LOQ) to 400 μg/kg for doxycycline, and the calibration graphs for tetracycline, chlortetracycline, oxytetracycline, their 4-epimer derivatives and tigecycline showed a good linear relationship within the concentration range from the LOQ to 200 μg/kg. The limits of detection (LODs) for the eight targets were in the range of 0.06 to 0.09 μg/kg, and the recoveries from the fortified blank samples were in the range of 89% to 98%. The within-run precision and between-run precision, which were expressed as the relative standard deviations, were less than 5.0% and 6.9%, respectively. The applicability was successfully demonstrated through the determination of residues in 72 commercial chicken samples purchased from different sources. This approach provides a novel option for the detection of residues in animal-derived food safety monitoring.

## 1. Introduction

The efficacy of tetracycline drugs in treatment, prevention and feed supplementation has been examined in long-term situations and widely authenticated. China’s current national food safety standards include specifications only for tetracycline (TC), chlortetracycline (CTC), oxytetracycline (OTC) and doxycycline (DOXY) among tetracycline drugs, and the maximum residue limit (MRL) for DOXY in poultry muscle is 200 μg/kg, whereas the MRLs for the other three substances in poultry muscle are 100 μg/kg [1]. The MRLs of the four tetracyclines stipulated by the European Medicines Agency are all 100 μg/kg [2]. Tetracyclines can specifically bind to position A of the 30S subunit of the bacterial ribosome, preventing it from binding to t-RNA on the ribosome and inhibiting the synthesis of peptide chains and proteins. The multiple active groups (2-acyl groups) contained in the structure of tetracyclines can bind strongly to proteins and, thus, can be widely distributed in body fluids and tissues after entering animal bodies [3]. However, tetracyclines have been used for decades since their successful introduction, causing intractable residual hazards and global clinical resistance problems. Possible health hazards include intestinal flora imbalance, calcium-chelated deposition, hepatotoxicity [3] and phototoxicity [4]. These hazards are specific to the treatment level and are not necessarily triggered at the residual level, but the potential consequences of residues that eventually accumulate in the human body through the food chain are unclear. Tetracyclines may contaminate water and soil due to discharge from animal husbandry and hospitals, insufficient treatment in wastewater treatment plants, and runoff flowing through manure or wastewater treatment sites and aquaculture areas [5]. The high resistance rate of microorganisms to tetracyclines [6] and the high incidence of tetracycline resistance genes in the environment have attracted much attention [7]. Therefore, the various concerns faced by traditional tetracyclines require the development and approval of novel, highly efficient and long-lasting tetracyclines.

Tigecycline (Synonyms: GAR-936, TGC) is a glycylcycline synthesized by adding a tert-butyl-glycylamido side chain to C9 of the D ring of the tetracycline main chain of semisynthetic tetracycline-minocycline, which belongs to a new broad-spectrum antibacterial tetracycline drug. The bacteriostatic mechanism of tetracyclines is to bind to the helical region (H34) on the 30S subunit of the ribosome, which prevents the transfer RNA from binding to the A binding site and, thus, prevents the formation of peptide chains [8]. The binding efficiency of TGC to the high-affinity site on the ribosomal subunits was estimated to be 5 times higher than that of TC [9]. Tetracyclines can not only be photoactivated under illumination and eradicate bacteria but also inhibit ribosomes to prevent bacterial regrowth [10]. The low absorption rate of TGC after oral administration [11] and the poor efficacy in the treatment of gastrointestinal infections except in turkeys after oral administration [12] may be important reasons explaining why TGC is not widely used in veterinary medicine, but many reports investigate TGC resistance. Chromosome-encoded efflux pump-mediated and plasmid-mediated variants of the resistance gene tet(X) cause TGC resistance to rapidly become prevalent [13]. The resistance rate of *Escherichia coli* isolated from the cecal contents of organically raised broilers in Austria to TGC is 2.0–27.6%, and the result is surprising because the broilers that were investigated had never received any tetracyclines at any point in their lives [14]. Among fecal samples collected from 157 livestock farms in four Chinese provinces, TGC resistance genes in chicken feces were more abundant than those in pig and cattle feces [15]. The treatment of carbapenem-resistant infections is usually a synergistic combination of TGC, polymyxins and aminoglycosides, but coresistance to this drug combination of carbapenem-resistant gram-negative bacteria has been increasingly reported worldwide [16]. Herein, the spread of TGC resistance and its potential health hazards, such as hepatotoxicity, severe hypoglycemia in diabetic patients and pancreatitis, mean that a reliable method for the simultaneous determination of TGC and other tetracyclines in animal-derived foods is needed.

The reported methods for the determination of TGC include rapid turbidimetric assay [17], monoclonal antibody-based immunochromatography [18], liquid chromatography (LC) [19,20,21,22,23,24] and LC–tandem mass spectrometry (LC–MS/MS) [25,26,27,28,29,30,31,32,33,34,35,36]. Two published works use LC–diode array detection (LC–DAD) as an LC method [19,20], and another four use LC–ultraviolet detection (LC–UVD) [21,22,23,24]. LC–MS/MS methods have been developed in various samples, including human serum [25], human plasma [26,27,28], turkey plasma [29], human plasma and cerebrospinal fluid [30], human bone [31], rat bone [32], human skin [33], human lung epithelial cells and polymorphonuclear neutrophils [34], human gall bladder, bile, colon, bone, synovial fluid, lung and cerebrospinal fluid [35], and rat brain tissues [36]. Overall, there is no research on the simultaneous detection of TGC and other tetracyclines in animal-derived foods using LC–MS/MS methods. In the present study, a novel and sensitive LC–MS/MS method was developed, optimized and validated for the concurrent determination of TGC, four tetracyclines (TC, CTC, OTC and DOXY) and their three 4-epimer derivatives (4-epi-TC, 4-epi-CTC and 4-epi-OTC) in chicken muscle. Our proposed approach to include a broader range of targets is a technological safeguard enhancement for food safety.

## 2. Results and Discussion

### 2.1. Stability of Standard Solutions

Working solutions were reconstituted after one week of use, and stock solutions were reformulated after one-month intervals. The degradation pathways of tetracyclines mainly involve three reactions: hydroxylation, dealkylation and deamination [37]. The target stability in the solution and matrix can be influenced by factors such as pH, microorganisms, metal ions, dissolved organic matter, light and temperature [3]. Under weakly acidic conditions, CTC is most prone to degradation because of the elimination reaction between the hydroxyl group at the C6 position and the hydrogen at the trans C5 position [38]. The degradation rate of tetracyclines accelerated with increasing temperature, and OTC was the most sensitive to increasing temperature [3]. Chelation with metal ions and microbial metabolic activity can accelerate the decomposition of tetracyclines [3,37]. Whether dissolved organic matter promotes or inhibits the photolysis of tetracyclines depends on the form, concentration and photochemical conversion of the organic matter itself and on factors such as pH and light intensity [3,4]. Overall, the preparation and storage of the stock and working solution should be kept at low temperature and away from light. A weak acid solution was used to dilute the working solution; therefore, as much as possible, the solution was freshly prepared before use.

### 2.2. Optimization of Sample Preparation

The experiment was carried out in accordance with the relevant regulations of the EU Commission Decision 2002/657/EC [39] and the U.S. Food and Drug Administration Bioanalytical Method Validation 05/24/18 [40]. As stated in the introduction, the sample objects for TGC determination are blood and tissue [25,26,27,28,29,30,31,32,33,34,35,36]. The method of simultaneously determining tetracyclines, including TGC, established in this study should not be compared with other determination studies involving TGC [25,26,27,28,29,30,31,32,33,34,35,36] in terms of sample preparation. Since solvent extraction was operated as the extraction method of targets, this study is comparable with the study that used solvent extraction for sample preparation in the simultaneous detection method for tetracyclines in muscle (or meat), as ascertained in Table 1. Some studies did not perform purification; the appropriate extractant was selected in the solvent extraction process, and the difference in solubility was sufficient for extraction and purification. In this study, a 0.1 mol/L Na_2_EDTA–McIlvaine buffer solution (pH 4.0), hydrophilic–lipophilic balance (HLB) column and nitrogen blowing were used for extraction, purification and concentration, respectively. After the simple and effective sample preparation method, the impurity interference in the sample is slight, and the recovery is ideal.

Factors such as extraction efficiency, time cost, solvent cost and toxicity, deproteinization and sample permeability should be comprehensively considered in sample preparation. Tetracyclines are amphoteric compounds; the acidic groups include phenolic hydroxyl and enol hydroxyl groups, and the basic group is dimethylamino. Tetracyclines have high solubility in alcohols, are easily degraded in both strong acid and alkaline solutions and are relatively stable in slightly acidic solutions. Under strong acid conditions with pH < 2, the secondary hydroxyl group at the C6 position and the hydrogen at the C5a position of tetracyclines are prone to elimination reactions due to the trans configuration and, thus, degradation. Under alkaline conditions, the hydroxyl group at the C6 position easily combines with the carbonyl group at the C11 position to be isomerized. Therefore, methanol was used to dissolve the standard in the preparation of the stock solution, a slightly acidic solvent was used as the diluent in the preparation of the working solution, and trichloroacetic acid was used in both diluent and mobile phase B to reduce possible solvent interference. Acidified protein precipitants (methanol and acetonitrile) [41,42,47,49] and buffer solution [43,44,45,46,48,50] were used as extractants, and acidifiers included formic acid, acetic acid, trifluoroacetic acid and EDTA. EDTA was used as the extractant in this study to minimize the chelation of tetracyclines with metal cations in biological samples while maintaining the freeness of the targets [41]. On this basis, after screening and comparing the extractants stated in Table 1, Na_2_EDTA–McIlvaine buffer solution was ultimately used as the extractant, which could not only extract the targets effectively but could also ensure that the targets were effectively retained in the SPE column filler, and the recoveries of the eight targets were above 89%.

Solid-phase extraction (SPE) is a column chromatographic separation process. Factors such as the fillers and specifications of the column and the property and concentration of the targets can affect the purification and enrichment efficiency. In the current consensus, the advantage is that various types of SPE columns are successfully commercialized and easily obtained, emulsification does not easily occur in the process, and the recovery rate is high. In contrast, the disadvantage is that biological samples are easy to block and have poor repeatability. The specific steps include pretreatment (activation and balance), sampling, leaching the interference components and eluting and recovering the targets. Independent solvent extraction can obtain sufficient recovery [42,48], but eliminating matrix interference is insufficient, so the superior purification ability of SPE is necessary. The extraction mechanisms in the publications stated in Table 1 include weak anion exchange (Strata-XL column) and nonpolar (C18 column) and polar (HLB column and Bond Elut ENV column) mechanisms. The separation and purification effects of these columns were compared. The Strata-XL column was not good for target retention, the Bond Elut Env column had a high flow rate but poor purification effect and the HLB column was more resistant to desiccation than the C18 column under the same specifications. As such, an HLB column with appropriate specifications was selected.

### 2.3. Optimization of HPLC–MS/MS

Among the reported TGC detection methods [17,18,19,20,21,22,23,24,25,26,27,28,29,30,31,32,33,34,35,36], there is no method for the simultaneous determination of TGC and other tetracyclines in animal-derived foods. The analytes in this study included four tetracyclines (TC, CTC, OTC and DOXY), as specified in China’s current national food safety standards, coupled with their three 4-epimer derivatives (4-epi-TC, 4-epi-CTC and 4-epi-OTC) and the novel glycylcycline (TGC). Of the studies where HPLC–MS/MS was used as the detection method, some studies [41,42,43,44,45,46] investigated a range of analytes that was not as comprehensive as that used in the present study. The analytes involved in the study by Nakazawa et al. [48] included demeclocycline but did not contain 4-epimer derivatives or TGC. The analytes reported in Desmarchelier et al. [47] are partially identical to those investigated in the present study.

To avoid isomerization and degradation, slightly acidic solvents were used in the LC separation of tetracyclines, such as formic acid [41,42,43,44,45], oxalic acid [43,46,47,48,49] and malonic acid [50]. In the study published by Tölgyesi et al., a LC–MS/MS method and a LC–DAD method were established, and formic acid and oxalic acid were used in the mobile phase, respectively [43]. In the mobile phase, trichloroacetic acid competes for charge, which affects the ionization efficiency of the targets. The ion suppression effect of the negative ion scanning mode is particularly obvious. The ionization of the targets in this study was carried out by positive ion scanning, and the SunFire column, which uses bonding and end-blocking techniques, is remarkably stable at low pH, allowing the low-concentration and non-volatile trichloroacetic acid to perform peak-shape improvements. Additionally, Bayliss et al. proposed that trichloroacetic acid is resistant to the epimerization and degradation of tetracyclines [51]. The different compositions of solvents and different concentrations of additives in the mobile phase were compared and evaluated. It was found that better target resolution and sensitivity were obtained with methanol as the organic solvent. Furthermore, the solubility of phospholipids in methanol is higher than that in acetonitrile, and methanol was used as the organic phase to reduce the matrix effect (ME) caused by the residual phospholipids. In addition, the setting for the gradient program was optimized to ensure that the targets had adequate resolutions. Both the present study and the report by Nakazawa et al. [48] used a C8 reversed-phase column to retain the targets, but the interface collocated in the latter study was an atmospheric pressure chemical ionization (APCI) source. Generally, the difference in ionization methods makes electrospray ionization (ESI) more sensitive to MEs than APCI. In addition to MS/MS detection, fluorescence detection (FLD), DAD and UVD have also been proposed. In research establishing the MS/MS method and DAD method, the LODs and LOQs of the latter are higher, and the recovery rate is lower [43]. Another report establishing the DAD method also has poor sensitivity [49]. Compared to LC methods [43,49,50] without qualitative capabilities, LC–MS/MS methods offer distinct advantages in terms of being able to analyze more compounds simultaneously and achieving lower sensitivity (as assessed by LOD and LOQ). The analysis efficiency (eight analytes) and recovery of the simultaneous determination method are high.

The ion source was equipped with an ESI source, which applied high voltage to charged droplets to generate aerosols and then produced ions that were separated by a quadrupole mass analyzer using electromagnetic fields. The efficiency of the reversed-phase separation and the ionization efficiency of the concentration-sensitive ion source ESI need to be comprehensively considered. The appropriate ionization efficiency was obtained by reducing the flow rate (0.3 mL/min) and increasing the curtain gas pressure (25 psi) and ion source temperature (500 °C). Tetracyclines have a higher response in positive ion mode due to the presence of formamide groups. Electrospray positive ion scan (ESI+) mode was implemented to scan the first-order mass spectra of the 50 ng/mL MS tuning solution. The average value after 50 scans was calculated by the mass/charge ratio (*m*/*z*) of the precursor ions, which is [M + H]^+^, where M is the relative molecular weight. After the precursor ions were determined, the product ions were scanned to obtain the secondary mass spectra (Figure 1) formed by the superposition, and the two ions with the highest abundance and noninterference were selected as the product ions. After the monitoring ion pair was selected, the multiple reaction monitoring (MRM) mode was utilized to optimize various mass spectrometry parameters.

### 2.4. Analytical Method Validation

The chromatographic peak area (y) of each quantitative ion pair (listed in Table 2) of the eight targets has a good linear relationship with the concentration gradient (x), and the correlation coefficient (R^2^) values are higher than 0.9997. The linear regression equations, R^2^ values and linear ranges are shown in Table 3.

Four quality control levels, LOQ, low-range, mid-range and high-range, were fortified into the blank sample and analyzed using the established method. The ratio of the calculated concentration obtained by substituting the peak area into the matrix-matched calibration curve to the true fortified concentration is the recovery. The mean recoveries obtained ranged from 89% to 98% with standard deviations (SDs) lower than 4.3%, indicating the high recovery stability of the targets. Both the within-run precision and between-run precision were evaluated by the relative standard deviation (RSD), and the ranges of the two were 2.0–5.0% and 3.6–6.9%, respectively. The between-run RSD can also be performed over a longer period of time, at which time the obtained RSD values may be higher. The relevant data are detailed in Table 4 and are in compliance with the regulation of the EU Commission Decision 2002/657/EC [39]. As specified by the EU Commission Decision 2002/657/EC, the acceptable RSDs are lower than 15%, and the acceptable recoveries are between 80% and 110% for concentrations higher than 10 μg/kg. When the concentration was less than 10 μg/kg, the acceptable recovery was greater than 60%, and the acceptable RSD was less than 20%.

Sensitivity is closely related to many factors, such as electrospray ionization efficiency, ion transmission efficiency, ME, resolution, chromatographic separation efficiency and target properties. The LOD and LOQ of each target are adopted to measure the sensitivity of the method, and they are expressed as the sample concentration required for the target to generate ion peaks with a certain signal-to-noise (S/N) ratio under sufficient resolution. The external fortified concentration of the target when the value of S/N ≥ 3 is the LOD, and the concentration when the value of S/N ≥ 10 is the LOQ. The LOD is defined as the lowest detectable concentration in the sample, and the LOQ is defined as the lowest concentration in the sample that can be quantified. The LOD and LOQ values obtained are shown in Table 5. Figure 2 shows the MRM mode’s overlayed extracted ion chromatogram and extracted ion chromatograms of the blank muscle samples fortified at 50 μg/kg.

The calculated decision limit (CCα) and detection capability (CCβ) values are presented in Table 5, and the values of CCα and CCβ are close to the MRLs of each target in chicken muscle, which meets the requirements of parameter validation of residue analysis in the EU Commission Decision 2002/657/EC [39] and the U.S. Food and Drug Administration Bioanalytical Method Validation 05/24/18 [40].

MS/MS is known for its powerful specificity; however, matrices, which are components other than targets in the sample, directly interfere with components of the analysis results, such as linearity, accuracy, precision and sensitivity. Endogenous components, such as phospholipids, can inhibit the ionization efficiency of the electrospray interface [52], and MEs can be caused by the introduction of exogenous components, such as plastic residues, SPE column fillers and buffer solutions, during sample preparation processing. Moreover, the ionization mode of the ionization source and the design of equipment manufacturers can also trigger MEs. In this study, MEs were estimated using the following formula:ME (%) = [(Slope _matrix-matched calibration curve_/Slope _solvent standard curve_) − 1] × 100%

The calculated ME values of TC, OTC, CTC, DOXY, TGC, 4-epi-TC, 4-epi-CTC and 4-epi-OTC were −25.3%, −27.2%, −24.9%, −43.6%, −35.2%, −23.2%, −25.7% and −20.9%, respectively. Positive and negative values correspond to ion enhancement and ion suppression, respectively. The range of −50% to −20% and 20% to 50% indicates a medium matrix effect [53]. All targets have medium ion suppression effects, which may be because some water-soluble proteins have not been effectively removed and compete with the targets for H+ during the ionization process. Various strategies have been applied in the effort to overcome the matrix effect, such as using SPE to remove phospholipids effectively in sample preparation, using ultrapure water to wash off phosphoric acid impurities in the SPE process, improving LC separation conditions, and selecting the appropriate injection volume and flow rate.

Applicability is an easily overlooked method validation parameter, and it must be evaluated with real samples to verify the real feasibility and reliability of the established method. Based on the established method, 72 samples were subjected to residue extraction and analysis, and only one refrigerated sample was found in which the OTC residue was higher than the regulatory limit. The newly developed method can be used as an available, authentic and stable method for the detection of TGC, tetracyclines and their 4-epimer derivatives in animal-derived muscle. Unlike tetracyclines, which are measured to ensure compliance with the corresponding MRLs, TGC is included to detect illegal use.

## 3. Materials and Methods

### 3.1. Chemicals and Reagents

TC (≥98.0% purity, CAS No. 60-54-8), CTC (≥99.0% purity, CAS No. 57-62-5) and OTC (≥97.0% purity, CAS No. 79-57-2) standards were purchased from Sigma-Aldrich LLC (St. Louis, MO, USA). DOXY (≥98.0% purity, CAS No. 564-25) and TGC (≥98.0% purity, CAS No. 220620-09-7) standards were obtained from Merck Drugs & Biotechnology Co., Inc. (Fairfield, OH, USA). 4-epi-TC (≥98.0% purity, CAS No. 101342-45-4), 4-epi-CTC (≥97.0% purity, CAS No. 23313-80-6) and 4-epi-OTC (≥97.0% purity, CAS No. 14206-58-7) standards were acquired from Acros Organics (Geel, Antwerp, Belgium), part of Thermo Fisher Scientific Corp. (Waltham, MA, USA).

Methanol and acetonitrile were of HPLC grade and were purchased from Tedia Co., Inc. (Fairfield, OH, USA). Ethyl acetate was provided by Sinopharm Chemical Reagent Co., Ltd. (Shanghai, China). Disodium ethylenediaminetetraacetic acetate (Na_2_EDTA, ≥99.0% purity), trichloroacetic acid (≥99.5% purity), citric acid (≥99.5% purity) and hydrogen sodium phosphate (≥99.0% purity) were obtained from Solarbio Life Science Co., Ltd. (Beijing, China). Ultrapure water (18.2 MΩ*cm, 25 °C) was prepared in real time by a Milli-Q HR 7000 intelligent water purification system (Merck Drugs & Biotechnology Co., Inc., Fairfield, OH, USA). All solutions injected into the LC separation system were degassed by a P300H ultrasonic degasser (Elma Electronic GmbH, Pforzheim, Stuttgart, Germany). Na_2_EDTA (37.18 g), citric acid (12.9 g) and hydrogen sodium phosphate (10.9 g) were dissolved in 1 L of ultrapure water to prepare a 0.1 mol/L Na_2_EDTA–McIlvaine buffer solution.

### 3.2. Preparation of Stock and Working Solutions

Eight standards were appropriately weighed to convert the purity to 100% and were placed into 10 mL brown volumetric flasks. The standards were dissolved in 10 mL methanol to prepare 8 stock solutions of 1.00 mg/mL. The stock solutions were prepared fresh each month and stored in an ultralow temperature refrigerator (−18 °C). The working solutions were prepared by diluting the corresponding stock solution gradually with methanol:0.01 mol/L trichloroacetic acid aqueous solution (5:95, V:V). They were stored at 4 °C, protected from light and freshly prepared every week.

### 3.3. LC and MS/MS Conditions

LC separation was performed using an Alliance e2695 HPLC system (Waters Corp., Milford, MA, USA). The analytes with an injection volume of 30 µL were separated and retained on a Waters SunFire C8 reversed-phase column (50 mm × 2.1 mm) with a particle size of 5 µm. The column temperature was controlled by a thermostat at 28 °C. Methanol (A) and 0.01 mol/L trichloroacetic acid aqueous solution (B) constituted the mobile phase and were pumped into the HPLC system at a constant flow rate of 0.3 mL/min. The gradient elution program was as follows: 0, 95% B; 3 min, 70% B; 6 min, 66.5% B; 8 min, 35% B; 12 min, 35% B; and 15 min, 95% B.

A triple quadrupole tandem mass spectrometer (Triple Quad 5500, Applied Biosystems Corp., Framingham, MA, USA) was installed with real-time monitoring, acquisition and data analysis software named Analyst (Applied Biosystems Corp., Framingham, MA, USA). The ion source temperature was set to 500 °C. Mass spectrometry was performed in positive ion scanning and MRM modes. The ionization voltage, collision chamber outlet voltage and injection voltage were set at 4500 V, 15 V and 10 V, respectively. The spray gas pressure, auxiliary heating gas pressure, curtain gas pressure and collision gas pressure were set as 50 psi, 60 psi, 25 psi and 8 psi, respectively. The dwell time of each quantitative and qualitative ion pair was set to 100 ms. The relative molecular masses of the eight tetracyclines and optimized mass spectrum parameters are listed in Table 2. Declustering potential removes solvent clusters, and collision energy breaks ions, both of which have optimal values and can be automatically and repeatedly optimized. Excessive declustering potential leads to fragmentation in the ion source, and excessive collision energy leads to continuous fragmentation of product ions and low responses.

### 3.4. Animals and Sample Preparation

This study was approved by the Institutional Animal Care and Use Committee of Jiangsu Province (Permit Number 45) and the ethics committee of Yangzhou University, and this study was conducted in accordance with the Declaration of Helsinki. All the experimental processes related to animal feeding and slaughter were carried out in strict accordance with the recommendations of the Guide for Ethical Review of Laboratory Animal Welfare issued by the National Technical Committee for Standardization of Laboratory Animals and the Guide for the Care and Use of Laboratory Animals of Jiangsu Province. All efforts were made to minimize the suffering experienced by the animals. Ten-week-old Jinghai yellow chickens were used, with 30 roosters and 30 hens each. The males and females were divided into flocks and raised on the ground covered with padding at a rearing density of 5 per/m^2^. The experimental birds were maintained under controlled environmental conditions (21 ± 3 °C temperature, 50–60% relative humidity, natural light). No antibiotics were provided in the feed or drinking water, and no antibiotic treatment was administered. After 2 weeks of rearing, 10 chickens were randomly selected from the rooster flock and the hen flock for sacrifice. The chickens were anesthetized with sodium pentobarbital and sacrificed immediately by manual exsanguination. The left breast muscle of each chicken was collected, chopped, homogenized and stored as the blank sample in a refrigerator at −18 °C.

The blank sample homogenate (2.0 ± 0.02 g) that was restored to room temperature was accurately weighed by a fully automatically calibrated electronic balance (ME204T/02, Mettler Toledo Co, Ltd., Zurich, Switzerland) and then loaded into capped 50 mL conical-bottom centrifuge tubes. Subsequently, 10 mL of 0.1 mol/L Na_2_EDTA–McIlvaine buffer solution (pH 4.0) was poured in for extraction. After 2 min of vortex vibration and 10 min of high-frequency ultrasonic oscillation, the sample solution was centrifuged for 5 min with the centrifugal acceleration set to 3000× *g* and the temperature set to 4 °C. The residue was repeatedly extracted, and the combined supernatant was vortex-mixed for 3 min after the addition of 10.0 mL n-hexane saturated by acetonitrile and then centrifuged at 5000× *g* and 4 °C for 5 min. After the upper layer of fat was suctioned out, the final supernatant was filtered through filter paper (1005-150 grade 1, Whatman Co., Ltd., Metstone, England, UK) for further purification and concentration.

Methanol (3 mL) and ultrapure water (3 mL) in the proper sequence were injected into a 60 mg/3 mL column (Oasis HLB, Waters Corp., Milford, MA, USA) for activation and equilibrium. Immediately afterward, the extract was added when the liquid in the barrel of the column flowed cleanly, but the filler was still saturated. In the sample loading step of the SPE process, a sealed vacuum pump was connected to assist in controlling the flow smoothness and flow rate (≤1 mL/min). Then, the column was washed with 3 mL of ultrapure water and 3 mL of methanol:ultrapure water (5:95, V:V). Ultimately, 10 mL of methanol:ethyl acetate with a volume ratio of 1:9 was pipetted for elution. The collected eluate was evaporated to near dryness at 40 °C under gentle and steady nitrogen flow. The nearly dry precipitate was reconstituted with 1 mL of initial mobile phase, vortexed for 30 s and filtered through a 0.45 μm × 13 mm sterile membrane filter (PTFE, ANPEL Laboratory Technologies (Shanghai) Inc. Shanghai, China). Aliquots were transferred to brown sampler vials until 30 μL was drawn into the HPLC system by an automatic injection needle for analysis.

### 3.5. Analytical Method Validation

The method established in this study was validated in accordance with the guidelines described in the EU Commission Decision 2002/657/EC [39] and the U.S. Food and Drug Administration Bioanalytical Method Validation 05/24/18 [40]. The external standard method was applied for quantification, and the matrix-matched calibration curve, which is suitable for on-site detection of a large number of samples, was drawn to calculate the residue.

The matrix-matched calibration curve was a concentration–response regression model evaluated with seven calibrated concentrations for each target starting from the corresponding LOQ. Blank chicken muscle was treated according to the established sample preparation method to obtain the blank matrix solution, and then the standard working solutions were fortified. The calibrated concentration gradient set based on the MRL for DOXY was LOQ, 5, 25, 50, 100, 200 and 400 μg/kg, and the calibrated concentration gradients for other targets were LOQ, 5, 10, 25, 50, 100, 150 and 200 μg/kg. In each validation run, the calibrator level of the LOQ met ±20% of the theoretical concentrations, other calibrator levels met ±15% of the theoretical concentrations, and a minimum of six calibrator levels met these criteria [39].

An appropriate amount of working solution was fortified into the blank sample so that the true fortified concentration of the target was exactly at the 4 quality control levels: LOQ, low-range (3 LOQ), mid-range (MRL) and high-range (2 MRL), with 6 parallel samples for each level [40]. After the sample preparation and LC–MS/MS detection described above, the peak area of the quantitative ion pair of each target was brought into the corresponding matrix-matched calibration curve, and the recovery was calculated from the ratio of the measured concentration to the true fortified concentration.

The same equipment and the same matrix-matched calibration curve were used by the same operator in the same laboratory for independent runs of blank samples fortified to the four quality control levels of LOQ, low-range, mid-range and high-range in one day, and each level contained 6 parallel samples. The ratio of the SD of the obtained recovery data to the average recovery is the RSD [39] or coefficient of variation (CV) [40] for the evaluation of within-run precision (repeatability). Four quality control levels were measured 3 days in one week, each level included 6 parallel samples, and the matrix-matched calibration curves were plotted daily within 3 days. The obtained RSD was used to evaluate between-run precision (reproducibility within the laboratory). Reproducibility means that precision was obtained by different operators using the same method and different equipment in different laboratories [39]. The impact of carryover on the accuracy of the sample concentrations was assessed, and carryover did not exceed 20% of the LOQ, thus meeting the guidelines of the U.S. Food and Drug Administration [40].

The sensitivity was assessed by the LOD and LOQ calculated by the S/N ratio method. The methods to calculate the S/N ratio include the peak-to-peak S/N ratio, half peak-to-peak S/N ratio and root mean square S/N ratio, and the peak-to-peak S/N ratio method was used for calculation in this study. When the average S/N ratio of the product ions after 6 parallel samples ≥3, the true fortified concentration was the LOD. The LOQ referred to the true fortified concentration when the average S/N ≥ 10. The acceptance criterion for sensitivity is that the precision should be ±20% CV [39,40].

The MEs of the LC–MS/MS method are due to the direct influence of the coeluting components of the targets on the ionization efficiency; thus, it exists objectively and manifests as ion enhancement and suppression. We calculated the ME by comparing the slope of the matrix-matched calibration curve with the slope of the solvent standard curve [53].

CCα was calculated as MRL + 1.64 × SD (α = 5%), and CCβ was calculated as CCα + 1.64 × SD (β = 5%) [39]. The SD was calculated from 20 blank chicken muscle samples fortified with the corresponding MRL for each target according to the method for evaluating the precision under reproducibility within the laboratory conditions described in Section 2.4.

To test the applicability of the method to its intended purpose, the development team purchased 72 pieces of chicken breast belonging to different brands that utilized different storage methods, such as freezing (−18 °C), chilling (approximately 0 °C) and refrigeration (0–4 °C), from local supermarkets, farmers’ markets and wholesale markets to determine the residues using the method established in this study.

## 4. Conclusions

TGC, tetracyclines (TC, CTC, OTC and DOXY) and their 4-epimer derivatives (4-epi-TC, 4-epi-CTC, 4-epi-OTC) can be precisely, sensitively and rapidly measured in chicken muscle through the implementation, optimization and validation of analytical methodologies. The simple sample preparation procedure consists of simple solvent extraction partitioning and subsequent SPE on an HLB column. The sample is ionized by an ESI source after being separated and retained in a reversed-phase LC system, and eight targets can then be concurrently identified and separately quantified. The recoveries of the established methods were all above 89% and the levels of sensitivity and precision were sufficiently low; CCα, CCβ and matrix effects were also assessed. The developed approach was favorably employed for the quantitative analysis of residues in real chicken samples, which demonstrates the applicability, fitness and feasibility of the method.

## Figures and Tables

**Figure 1 molecules-27-06139-f001:**
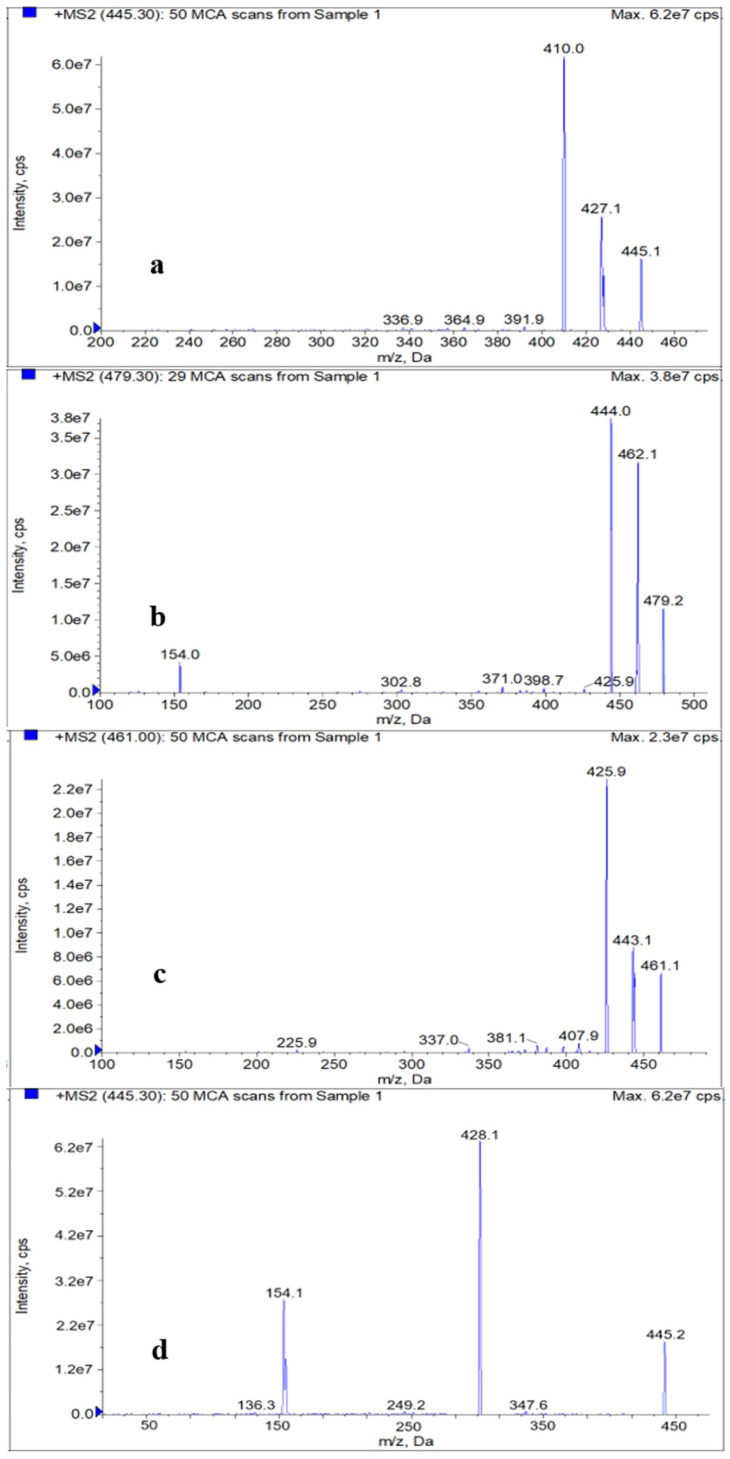
Tandem mass spectra of TC (**a**), CTC (**b**), OTC (**c**), DOXY (**d**), TGC (**e**), 4-epi-TC (**f**), 4-epi-CTC (**g**) and 4-epi-OTC (**h**).

**Figure 2 molecules-27-06139-f002:**
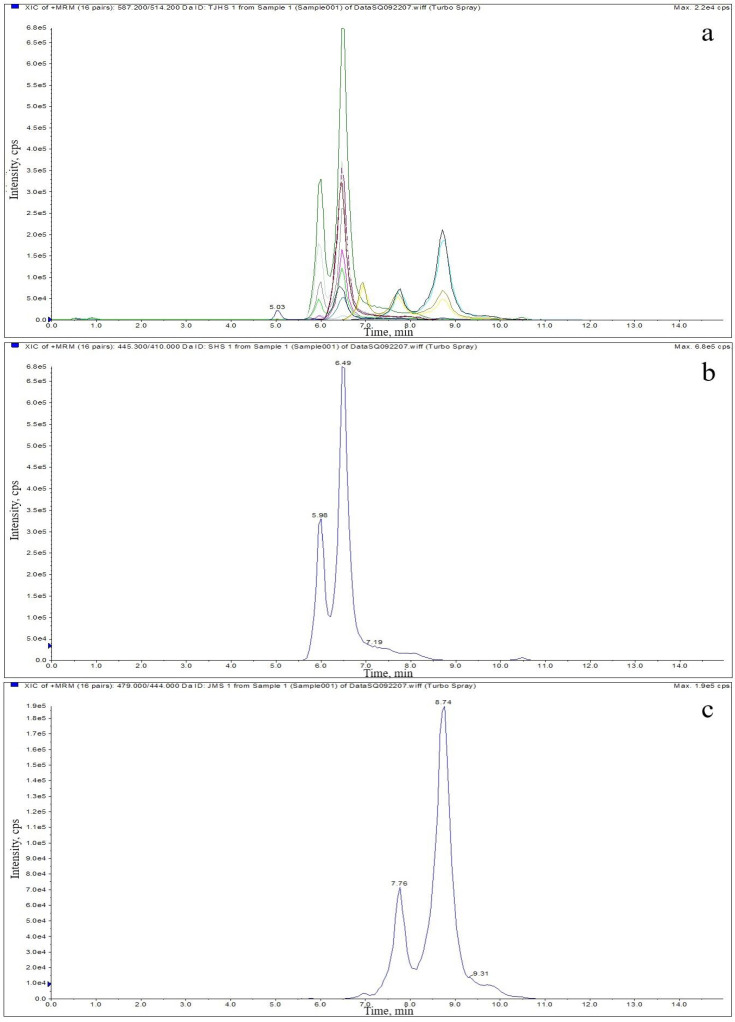
Overlayed extracted ion chromatogram (**a**) and extracted ion chromatograms of blank chicken muscle fortified with TC (**b**), CTC (**c**), OTC (**d**), DOXY (**e**), TGC (**f**), 4-epi-TC (**g**), 4-epi-CTC (**h**) and 4-epi-OTC (**i**) at a concentration of 50 μg/kg.

**Table 1 molecules-27-06139-t001:** Comparison of LC and LC–MS/MS methods for the simultaneous detection of tetracyclines in muscle (or meat) using solvent extraction for sample preparation.

Analytes	Sample Preparation	LC Conditions	Detection Method	Linearity Range (μg/kg)	Sensitivity (μg/kg)	Recovery (%)
CTC, OTC, DOXY (chicken) [41]	Extracted with acetonitrile containing 0.025 mol/L formic acid; Low-temperature cleanup	Mobile phase: water containing 0.025 mol/L formic acid–methanol containing 0.025 mol/L formic acid; stationary phase: C18 column (50 mm × 4.6 mm, 1.8 μm)	HPLC–MS/MS	20–400	LODs: 10.0; LOQs: 10.0	98.4–103.2
TC, CTC, OTC, DOXY (chicken) [42]	Extracted with methanol and 0.01 mol/L EDTA containing 1% formic acid	Mobile phase: water containing 0.1% formic acid–methanol containing 0.1% formic acid; stationary phase: C18 column (75 mm × 4.6 mm, 5 μm)	HPLC–MS/MS	50–200	LODs: 7.9-14.6; LOQs: 24.2–44.3	56.9–101.2
TC, CTC, OTC, DOXY (pig meat) [43]	Extracted with McIlvaine buffer and 2% acetic acid (pH 2.0); SPE using Strata-XL column	Mobile phase: water containing 0.1% formic acid–acetonitrile; stationary phase: C18 column (150 mm × 4.6 mm, 2.6 μm); Mobile phase: water containing 0.01 mol/L oxalic acid–acetonitrile; stationary phase: C18 column (150 mm × 4.6 mm, 2.7 μm)	HPLC–MS/MS; HPLC–DAD	25–200	LODs: 5.0–10.0, LOQs: 17.0-33.0; LODs: 0.5, LOQs: 1.7	79.9–106.7; 51.9–80.1
TC, CTC, OTC, DOXY (chicken) [44]	Extracted with 0.1 mol/L Na_2_EDTA–McIlvaine buffer; SPE using C18 column	Mobile phase: water containing 0.1% formic acid–acetonitrile containing 0.1% formic acid; stationary phase: C18 column (100 mm × 2.0 mm, 5 μm)	HPLC–MS/MS	0–200	LOQs: 7.0–35.0	89.4–106.3
CTC, OTC, 4-epi-CTC, 4-epi-OTC (chicken) [45]	Extracted with EDTA–McIlvaine; SPE using C18 column	Mobile phase: water containing 0.1% formic acid–methanol containing 0.1% formic acid; stationary phase: C18 column (150 mm × 2.1 mm, 3.5 μm)	HPLC–MS/MS	20–200	LODs: 20.0; LOQs: 21.2–21.6	94.0–108.0
TC, CTC, OTC, DOXY, 4-epi-TC, 4-epi-CTC, 4-epi-OTC (pig muscle) [46]	Extracted with 0.1 mol/L sodium succinate solution (adjusted to pH 4.0 with 10 mol/L NaOH); SPE using HLB column	Mobile phase: 3% tetrahydrofuran containing 0.001 mol/L oxalic acid and 0.5% formic acid–tetrahydrofuran; stationary phase: PLRP-S polymeric column (250 mm × 4.6 mm, 8 μm)	HPLC–MS/MS	0–1000	LODs: 0.5–4.5	-
TC, CTC, OTC, DOXY, 4-epi-TC, 4-epi-CTC, 4-epi-OTC, 6-epi-DOXY; demeclocycline, 4-epi-demeclocycline (poultry and pig muscle, fish) [47]	Extracted with 0.1 mol/L EDTA and acetonitrile	Mobile phase: water containing 0.01 mol/L oxalic acid–methanol containing 0.1% formic acid; stationary phase: HSS T3 column (100 mm × 2.1 mm, 1.8 μm)	HPLC–MS/MS	0–400	-	-
TC, CTC, OTC, DOXY, demeclocycline (chicken and fish) [48]	Extracted with 0.1 mol/L Na_2_EDTA–McIlvaine buffer (pH 4.0); SPE using Bond Elut Env column	Mobile phase: methanol:acetonitrile:0.005 mol/L oxalic acid (18:27:55, V:V:V); stationary phase: C8 column (250 mm × 4.6 mm, 5 μm)	HPLC–APCI/MS/MS	0–0.50 ppm	LODs: 0.001–0.004 ppm	60.1–88.9
TC, CTC, OTC (fish and shellfish) [49]	Extracted with 20% trifluoroacetic acid, EDTA and methanol:0.01 mol/L citrate (80:20, V:V, pH 4.0)	Mobile phase: 0.05 mol/L oxalic acid:acetonitrile:methanol (70:20:10, V:V:V); stationary phase: C18 column (250 mm × 4.6 mm, 3.5 μm)	HPLC–DAD	12.5–1250 (fish) 17.5–2500 (shellfish)	LODs: 15.0–62.0; LOQs: 125.0	95.0–105.0
TC, CTC, OTC, DOXY (chicken) [50]	Extracted with 0.1 mol/L citrate buffer (pH 5.0) and ethyl acetate; Solid-phase microextraction fiber based on molecularly imprinted polymers	Mobile phase: 0.1 mol/L malonate:0.05 mol/L magnesium chloride (30:70, V:V, adjusted to pH 6.5 with NH_3_·H_2_O); stationary phase: C18 column (250 mm × 4.6 mm, 5 μm)	HPLC–FLD	5–200 μg/L	LODs: 1.02–2.31 μg/L	72.6–92.8

- Not reported.

**Table 2 molecules-27-06139-t002:** Relative molecular mass and mass spectrum parameters of eight tetracyclines.

Analyte	Relative Molecular Mass	Transition (*m*/*z*)	Declustering Potential (V)	Collision Energy (eV)
TC	444.4	445.1 > 410.0 * 445.1 > 427.1	130 130	21 21
CTC	478.9	479.2 > 444.0 * 479.2 > 462.1	135 135	30 30
OTC	460.4	461.1 > 425.9 * 461.1 > 443.1	121 121	23 23
DOXY	444.4	445.2 > 428.1 * 445.2 > 154.1	131 131	31 31
TGC	585.7	587.2 > 514.1 * 587.2 > 457.3	120 120	28 28
4-epi-TC	480.9	445.1 > 410.0 * 445.1 > 427.1	130 130	21 21
4-epi-CTC	515.3	479.2 > 444.0 * 479.2 > 462.1	134 136	28 29
4-epi-OTC	460.4	461.1 > 425.9 * 461.1 > 443.1	121 121	20 21

* Quantitative ion pair.

**Table 3 molecules-27-06139-t003:** Linear regression equations, R^2^ values and linearity range of eight tetracyclines in chicken muscle.

Analyte	Regression Equation	R^2^	Linearity Range (μg/kg)
TC	y = 88,271 x + 8955.5	0.9999	0.15–200
CTC	y = 22,813 x + 71,883	0.9997	0.13–200
OTC	y = 30,672 x + 57,745	0.9998	0.16–200
DOXY	y = 19,943 x + 48,367	0.9997	0.12–400
TGC	y = 11,454 x + 32,725	0.9997	0.19–200
4-epi-TC	y = 150,673 x + 163,280	0.9998	0.17–200
4-epi-CTC	y = 48,531 x + 157,731	0.9998	0.16–200
4-epi-OTC	y = 39,020 x + 41,188	0.9999	0.18–200

**Table 4 molecules-27-06139-t004:** Recovery and precision of eight tetracyclines used to fortify blank chicken muscle (n = 6).

Analyte	Fortified Level (μg/kg)	Recovery (%)	RSD (%)	Within-Run RSD (%)	Between-Run RSD (%)
TC	0.15	90 ± 2.9	3.3	3.0	6.0
	0.45	92 ± 2.7	3.0	2.7	4.5
	100 ^α^	96 ± 3.3	3.4	3.8	5.9
	200	94 ± 3.7	4.0	3.5	5.0
CTC	0.13	89 ± 3.7	4.2	3.6	5.9
	0.39	93 ± 3.3	3.5	4.9	5.2
	100 ^α^	93 ± 2.3	2.5	3.7	5.7
	200	97 ± 3.5	3.6	3.2	6.9
OTC	0.16	91 ± 2.7	2.9	3.4	4.6
	0.48	95 ± 3.1	3.3	3.0	4.2
	100 ^α^	94 ± 4.0	4.1	3.4	4.8
	200	98 ± 4.3	4.4	4.0	5.9
DOXY	0.12	91 ± 2.1	2.3	3.2	5.7
	0.36	94 ± 3.2	3.4	4.3	5.9
	200 ^α^	95 ± 2.5	2.6	3.0	4.1
	400	92 ± 2.7	2.9	5.0	4.3
TGC	0.19	89 ± 2.9	3.2	3.6	5.2
	0.57	89 ± 2.6	3.0	3.8	5.8
	100 ^α^	93 ± 1.8	2.0	2.3	3.8
	200	94 ± 1.8	1.9	2.0	3.6
4-epi-TC	0.17	94 ± 2.6	2.8	3.7	5.7
	0.51	93 ± 3.1	3.3	3.2	5.9
	100 ^α^	90 ± 2.8	3.1	3.2	5.3
	200	92 ± 2.9	3.1	4.1	6.2
4-epi-CTC	0.16	93 ± 2.9	3.1	4.0	6.5
	0.48	96 ± 2.7	2.8	2.9	5.0
	100 ^α^	96 ± 1.8	1.9	2.1	3.6
	200	98 ± 2.1	2.1	3.1	4.5
4-epi-OTC	0.18	90 ± 3.1	3.5	3.8	6.4
	0.54	91 ± 2.2	2.4	3.0	4.0
	100 ^α^	94 ± 3.2	3.4	4.5	6.9
	200	94 ± 3.0	3.2	2.9	4.3

^α^ MRL.

**Table 5 molecules-27-06139-t005:** LODs, LOQs, CCα and CCβ of eight tetracyclines in chicken muscle.

Analyte	LOD (µg/kg)	LOQ (µg/kg)	CCα (µg/kg)	CCβ (µg/kg)
TC	0.09	0.15	104	108
CTC	0.06	0.13	119	129
OTC	0.08	0.16	104	109
DOXY	0.08	0.12	206	207
TGC	0.07	0.19	102	106
4-epi-TC	0.08	0.17	101	102
4-epi-CTC	0.06	0.16	102	104
4-epi-OTC	0.07	0.18	103	103

## Data Availability

All available data are contained within the article.

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
