# Peer review of "Concurrent Determination of Tigecycline, Tetracyclines and Their 4-Epimer Derivatives in Chicken Muscle Isolated from a Reversed-Phase Chromatography System Using Tandem Mass Spectrometry"

_molecules, 2022, doi:10.3390/molecules27196139_

Round 1

Reviewer 1 Report (New Reviewer)

This manuscript demonstrates a HPLC-MS/MS method in simultaneously determination tigecycline (TGC), tetracycline (TC), chlortetracycline (CTC), oxytetracycline (OTC), doxycycline (DOXY), 4-epitetracycline (4-epi-TC), 4-epichlortetracycline (4-epi-CTC) and 4-epioxytetracycline (4-epi-OTC) in chicken muscle samples. The sample preparation method and HPLC-MS/MS method were optimized and validated in the study. The developed method was also compared with the previous study in determining various tetracycline and tigecycline.

Comments:

-Tetracyclines and their epimers shared the same MRM and the peaks appear at the same retention time between both epimers. Apparently, LC-MS/MS method is not the method in differentiating and separating the epimers. What are the rationales in determining the epimers in this study?

-Since the maximum residue limit for tetracyclines and the derivatives are 100 – 200 mg/kg and the linearity of each analytes cover wide ranges. Why do the authors determine the LOQ levels down to 0.12 to 0.19 mg/kg? Any justification for the authors to detect analytes in low LOQ level?

- In Section 2.1, the authors discussed the stability of the standard solutions under different condition. Rationales in determining each parameter were justified and conclusion has been made, however, no scientific results presented. The reviewer suggests the authors demonstrate scientific results in the stability study.

-Table 1 demonstrated the comparison between previous study in simultaneously detecting tetracyclines in muscle samples. The reviewer suggests the author to include the current study in the table for comparison.

-In the abstract and Table 4, the linear range of the DOXY was reported as LOQ – 400 mg/kg, however, in Table 3, the linear range for DOXY is 0.12 – 200 mg/kg.

- In Table 4, the LOQ level reported for OTC and CTC is 0.16 mg/kg and 0.13 mg/kg, respectively, however, in Table 5, the LOQ level for OTC and CTC is 0.13 mg/kg and 0.16 mg/kg, respectively instead.

-Based on the s/n ratio, the LOQ is 3 times higher than the LOD. The reported LOQ of each analyte is approximately 2 times or less than 2 times higher than the LOD.

-The overlay extracted ion chromatogram (Figure 3a) is not clear. MRM for each analyte should be included in the overlay chromatogram. The authors demonstrated only MRM of TGC in the overlay chromatogram.

-The extracted ion chromatogram of DOXY (Figure 3e) shows the qualitative ion pair at transition of 445 > 154 instead of the selected quantitative ion pair at transition 445.2 > 428.1.

-The reviewer suggests the authors include sample chromatogram in the manuscript.

-L479-480: The authors conclude that the study uses a time-saving sample preparation procedure by using simple solvent extraction partitioning and SPE method, however, the steps described in section 3.4 requires at least 1 hour for the entire sample preparation procedure. Does the procedure more time-saving as compared to previous study?

-The title for Table 1 5th column, Table 3 4th column and Table 4 2nd column are not clear. 

Author Response

Thank you for your comments and suggestions on this manuscript. We comprehensively revised and improved the manuscript based on your professional, accurate and detailed advice. We hope that our responses satisfy you and that the revised version meets your requirements. All changes are marked in red.

Reviewer 2 Report (New Reviewer)

Molecules

molecules-1885997

Concurrent Determination of Tigecycline, Tetracyclines and Their 4-Epimer Derivatives in Chicken Muscle Isolated from a Reversed-phase Chromatography System Using Tandem Mass Spectrometry

Dear Editor,

The article deals with the development of a quantitative and qualitative method for analysis of e analysis of tigecycline, four tetracyclines and their three 4-epimer derivatives in chicken muscle. The topic is good and the manuscript has been well designed and written. It can be accepted after the necessary corrections are done. My comments and questions are below;

-    Give more information about these drugs and their effects in the introduction section

-                 Line 105: Which solution was used for the preparation of stock solutions? Same with the solvent in the lines 348 and 349?

-          Lines 245 and 246: Is this about the drugs?

-          Line 416: What kind of filter was used?

-          Conclusion section should be improved!

Author Response

Thank you for your comments and suggestions on this manuscript. We comprehensively revised and improved the manuscript based on your professional, accurate and detailed advice. We hope that all answers satisfy you and that the revised version meets your requirements. All changes are marked in red.

Reviewer 3 Report (New Reviewer)

Highlights and strengths of the manuscript are:

The known information about tetracyclines analysis precisely described. A thorough characterization of new method of analysis of 8 tetracyclines is described. The proposed assay is precise and sensitive and can be used to determine concentration of tetracyclines in chicken meat. The reference section is correct and includes recent relevant bibliography (but you must add dots at the end of abbreviation, like in J. Braz. Chem. Soc., not J Braz Chem Soc).

Specific comments and suggested revisions:

Sections 2.1 and 2.2. I don't get it, is this material a speculation or the results of experimental study. If the latter, it might be described as figure or table where experimental data should be described.

Figure 2. For better visual perception, the mass spectra should be congruent with the same m/z axis (the various ranges as 200-460, 100-500, 50-450 etc. makes it difficult to compare).

Table 3. The values of r2 are extremely high (0.9997-0.9999) for the matrix-matched calibration curve procedure. I can't find the description of matrix-matched calibration curve procedure in Section 3.5 so I can't estimate all sources of mistakes. The general sample preparation procedure includes buffer solution extraction, centrifugation, hexane/acetonitrile extraction, SPE extraction, evaporation, and redissolving. Each of these steps are the potential sources of errors. The values of r2 look like calibration curve was built without matrix-matching (solvent standard curve).

Figure 3. Why samples of pure TC, CTC, 4-epi-TC, and 4-epi-CTC gave two chromatographic peaks after ion extraction?

Line 275. The description of ME abbreviation should be added.

Line 280. The data of solvent standard curves should be added.

Line 299. The results of 72 samples analysis should be added.

Section 2.5. This section looks like a kind of discussion not comparison. It sums up the assays used for the analysis of tetracyclines without analysis of their advantages and disadvantages (which is necessary for comparison).

Possible suggested revisions:

In general, the paper is well written, but I have to emphasize the excess of the references to publications in Results and Discussion Section. I understand people wanting to highlight problem from all angles, but it was a little too much.

Author Response

Thank you for your encouragement. We appreciate the time and effort that you dedicated to providing feedback on our manuscript and are grateful for the insightful comments on and valuable improvements to our paper. Your comments make us feel that our efforts expended during experimentation and writing are worthwhile. We have critically reviewed the formatting of the references in accordance with the issues raised by the reviewer and have made changes, all of which are marked in red. Thank you again for your positive and constructive comments and suggestions on our manuscript.

Reviewer 4 Report (New Reviewer)

The authors aimed to develop a quantitative and qualitative method using a high-performance liquid chromatography  tandem mass spectrometry (HPLC–MS/MS) detection for  the analysis of tigecycline, four tetracyclines and their three 4-epimer derivatives in chicken muscle. The work has novelty and originality.

  The work is interesting, it is presented in a difficult manner to follow the procedures and results. The research design is well planned. The work was performed at a high scientific level. The present manuscript was found to be suitable to be published. However, there is a need to improve the quality of the manuscript.

Comments

1.     In Table 1 add a linearity range.

2.     Remove Fig. 1

3.     Greenness profile of analytical method should be added and compared with previously developed method.

4.     Stability study will add some impact to presented work.

5.     Choice of LC and MS/MS conditions should be discussed more.

6.     Fig. 2, 3 have bad quality.

7.     Section 3.1 Chemicals and reagents: Authors should indicate the identifying codes.

Author Response

We appreciate the time and effort that you dedicated to providing feedback on our manuscript and are grateful for the insightful comments on our work. Thank you very much for reviewing the manuscript and considering it to be novel, original, interesting, well planned and presented at a high scientific level. We have conducted a comprehensive revision of the manuscript and edited some unnecessary and ambiguous expressions, and we hope that these changes meet your standards. Thank you again for taking the time to review this manuscript.

Round 2

Reviewer 1 Report (New Reviewer)

The authors have satisfactorily responded to all my questions and made the necessary changes to the manuscript.

Reviewer 3 Report (New Reviewer)

The authors of the manuscript "Concurrent Determination of Tigecycline, Tetracyclines and Their 4-Epimer Derivatives in Chicken Muscle Isolated from a Reversed-phase Chromatography System Using Tandem Mass Spectrometry" have certainly heeded the comments made by the reviewer and corrected draft as much as possible. The manuscript can be accepted in present form. Good luck. 

This manuscript is a resubmission of an earlier submission. The following is a list of the peer review reports and author responses from that submission.

Round 1

Reviewer 1 Report

I have now completed my review on the manuscript: molecules-1758028, entitled “Concurrent Identification of Tigecycline, Tetracyclines and Their 4-Epimer Derivatives in Chicken Muscle Isolated from a Reversed-phase Chromatography System Using Tandem Mass Spectrometry”.

I think this is an interesting manuscript, providing a selective and sensitive method for the simultaneously analyses of tigecycline and tetracyclines, and their 4-epimer derivatives in chicken muscle.

General comments

I think you should improve the English writing, sometimes the sentences/paragraphs are written so that it is difficult for the readers to understand what it means.

Specific comments

1.     Line 29: To many numbers in the recovery, use 89 and 98 rather than 88.9 and 97.6

2.     Line 31: The within-run RSD is higher than 2.0 according to table 4.

3.     Line 59-60: I do not understand what this sentence means, and how it is related to the rest of the paragraph.

4.     Line 67: Which food safety studies have been performed before the use was extended to veterinary use. In which countries is it allowed to use for food producing animals, and is there a MRL established for it anywhere?

5.     In the EU where no MRL has been established, TGC is not allowed to be used, and the aim of the sampling will therefore be to detect illegal use, while for the tetracycline the aim of the sampling is to assure compliance with the MRL. Can you include in the discussion some thought about this?

6.     Paragraph 2.1:is the stability studies done in your lab, then include the data on this. If it is from the literature include references.

7.     Line 167-171: It is not clear here whether you decided on extractant based on data from table 1. or if you tested this yourself in the lab.  Please write this clearer. The same goes for the selection of SPE-column, and mobile phase.

8.     Paragraph 3.3: Could you include how the declustering potential and collision energy were optimized.

9.     Line 230: Do you know which part of the molecule is gone for the product ions?

10.  Line 270: The LOD is very low, and often with these low concentrations there could be problems with contamination/carryover from other samples, solutions, was this a problem with your method?

11.  Line 310: how many samples did you detect any of analytes in, could you make a table of the number of samples and range you detected each analyte in.  

12.  Line 396: use mL not milliliter, like elsewhere in the manuscript.

13.  Line 412: You have not included an internal standard in this method, why not? I am also very surprised that you get so low RSD without an internal standard.

14.  Line 426: Maybe samples is a better word than experiments. Also is these six parallels from the same batch, or different batches.

15.  Line 437: Preferably the between day precision runs should be performed over a longer time period than one week, so maybe add in the discussion that over a longer time period, the RSD might be higher.

16.  Line 456: Is these 20 samples from the same chicken matrix, or from 20 different chicken matrices.

17.  Table 2: Several of the analytes have the same mass transitions, could you elaborate on that. Was there some overlap of the different analytes in the retention time?

18.  Table 4: It is difficult to see which numbers belong to which analyte? Maybe put the analyte name on top of the cell, not in the middle, that way it could be easier to link the numbers to the analyte. Or use lines like in table 1.

19.  Table 5: Remover numbers after the period for CCa and CCb, so 104 instead of 103.62.

20.  Figure 2: In several of the panels it says 50 scans, but for some of them it is fewer, do you have the spectra after 50 scans for them as well?

21. have you participated in any proficiency test for this method? Or compared your results with another lab?

Author Response

We are grateful to the reviewer for considering this work interesting. Such comments make us feel that our efforts in experimentation and writing are worthwhile. We are also very grateful for your suggestions, which are professional and addressed key points.

In response to the reviewers’ detailed comments and suggestions, we comprehensively revised the manuscript point by point. We improved many expressions with ineffective phrasing and modified confusing sentences/paragraphs with great attention to detail. To avoid misunderstandings due to language errors, we sent the revised manuscript to American Journal Experts for revision by a native English speaker. We carefully proofread the manuscript to minimize typographical, spelling, grammatical, and formatting errors.

Each change is marked in red font. Thank you again for your opinion, which encouraged us to continue our work.

Reviewer 2 Report

I would not recommend its publication due to the fact that there are many articles to show efficient method for the analysis of such antibiotics. The other issue is that these residues are being analyzed using advanced LC-MS techniqes such as LC coupled high resolution mass spectrometry. 

Author Response

We appreciate the time and effort that you dedicated to providing feedback on our manuscript and are grateful for the insightful comments on our paper. We have also conducted a comprehensive review of the manuscript and edited some unnecessary and ambiguous expressions, and we hope that these changes meet your standards. The method we have developed is innovative in comparison to the methods reported, as demonstrated by our proposed title. We believe that what we have studied is complementary to existing detection methods (advanced high-resolution MS). Thank you again for taking the time to review this manuscript.

Reviewer 3 Report

The manuscript by Yawen Guo et al. is devoted to the development of a method for the determination of antibiotics in the muscles of chickens by HPLC-MS/MS. Ensuring the safety and quality control of food products is an important task. The authors set the goal of the study to develop a selective and sensitive technique that would ensure the simultaneous determination of a wide range of analytes. The manuscript has a number of serious issues to be clarified:

·         The Introduction is too long (information on analytes can be significantly reduced). At the same time, section 2 (2.1, 2.2 and others) often contains a lot of literature data that can be transferred to the introduction.

·         Section 2.2. I didn't see any information about the recovery provided by the extraction method chosen by the authors. Have these studies been carried out? This information (table) can be given in supplementary materials.

·         Figures 2 and 3 are too large, it is better to transfer them to supplementary materials. Table 2 can also be moved.

·         Figure 2. In the title of the figure, it is better to indicate that these are tandem mass spectra.

·         Line 183. At the expense of HLB, it is incorrectly stated that this is a non-polar phase.

·         Line 218. The authors are mistaken in believing that water in reverse phase chromatography is a strong eluent.

·         Line 222. Incorrect term «desolvent gas pressure».

·         Line 231. «multireaction monitoring» more correctly «multiple reaction monitoring».

·         Table 4. Excessive precision for Recovery and RSDs. Too many significant numbers.

·         The manuscript lacks a comparison of the achieved results (LOD, LOQ, etc.) with what is available in the literature. What are the advantages of what the authors offer?

·         The authors declare that they are developing an approach to the determination and identification of tigecycline, tetracyclines and their derivatives. What is identification in this case?

·         If real samples were studied in the work, then I would recommend that you provide in supplementary materials a chromatogram where analytes were found.

Based on this, the manuscript requires significant revision. I do not recommend accepting this work for publication in its present form.

Author Response

The reviewer provided us with professional, accurate and invaluable suggestions, clearly spending substantial time and effort reviewing our contribution. We are grateful for your hard work, which provided clear directions for modifying and improving our manuscript. We performed a comprehensive, point-by-point revision of the manuscript in accordance with the reviewer’s detailed comments and suggestions.

Reviewer 4 Report

The authors have presented a manuscript describing a SPE-LC-MS/MS method for the determination of antibiotics and their derivatives in chicken muscle sample.

The work was conducted in a competent way, and the manuscript was written in a scholarly way.

However there are some concerns that need to be addressed before the manuscript can be published:

1) the paper describes the quantitative role of the method, including linearity, LOQ, recovery etc. with lots of emphasis on the quantitative limits in the abstract, and yet the title says Identification. 

2) Please shorten the introduction, lines 62-80 can be reduced as they are not that relative to the manuscript

3) please explain in the introduction why chicken muscle was chosen as the sample

4) please include some results in section 2.2 when describing SPE (e.g. recovery)

5) Figure 3 a) this is not total ion chromatogram, this is overlayed extracted ion chromatogram

6) why are the authors using the word fortified instead of spiked?

7) please include a section with comparing the proposed method with already exhisting methods (e.g. those in Table 1). Emphasise why this method is novel and/or better. 

8) Include a table with the results of the 72 analysed samples (instead Figure 3 can go to Supplementary data)

9) The Conlusion needs to be completely rewritten. It does not reflect the content of the manuscript. Add the validation data and the results of the analysis 

Author Response

Thank you for your comments and suggestions on this manuscript. We comprehensively revised and improved the manuscript based on your professional, accurate and detailed views. All changes are marked in red font.

Round 2

Reviewer 2 Report

The authors made revision accordingly and the manuscript can be acceptable

Author Response

Thank you for your encouragement. Your comments make us feel that our efforts expended during experimentation and writing are worthwhile. We are grateful for your valuable counsel, which is professional and directed to key points. Thank you again for your positive and constructive comments and suggestions on our manuscript.

Reviewer 3 Report

The authors of the manuscript made the necessary corrections, reasonably answered all questions and comments. Significant technical errors have been fixed. I do not agree with some points, but they do not significantly affect the quality and importance of the study (for example, the use of a file with supplementary materials). I would like to clarify one more point:

Line 251. "within-run precision and between-run precision". It might be better to use the terms "intra-day precision and inter-day precision"

Table 4. "Recovery (%)" column. In this case, it is better to round off the error values.

After minor revision, the manuscript can be accepted for publication in the journal "Molecules".

Author Response

The authors of the manuscript made the necessary corrections, reasonably answered all questions and comments. Significant technical errors have been fixed. I do not agree with some points, but they do not significantly affect the quality and importance of the study (for example, the use of a file with supplementary materials). I would like to clarify one more point:

Answer: Thank you for your comments and suggestions on this manuscript. We comprehensively revised and improved the manuscript based on your professional, accurate and detailed review. We hope that our revisions will prove satisfactory and that the revised version meets your requirements. All changes are marked in red font.

Line 251. “within-run precision and between-run precision”. It might be better to use the terms “intra-day precision and inter-day precision”

Answer: Thank you for the professional advice. The terminology you suggested and the terminology we used are both correct and available. We expressed the precision in this way because we refer to FDA Bioanalytical Method Validation Guidance (screenshot 1). In addition, many manuscripts use a similar expression, as illustrated in screenshot 2 from DOI: 10.1016/s0731-7085(96)01951-6 and screenshot 3 from DOI: 10.1016/j.jchromb.2007.12.013.

 screenshot 1

 screenshot 2

 screenshot 3

We thus hope that you understand why we did not make the suggested changes. We apologize for disturbing you again and causing confusion to your review work.

Table 4. “Recovery (%)” column. In this case, it is better to round off the error values.

Answer: Based on the suggestions from other reviewers, the recovery column was previously changed to only two significant digits. If the SD values are rounded off, the adjacent columns also need to be modified. We thus have to keep the information as presented and hope that the reviewer will understand that we are not being disrespectful.

After minor revision, the manuscript can be accepted for publication in the journal “Molecules”.

Answer: We have thoroughly and carefully reviewed and improved the manuscript to meet the reviewer’s requirements, and we apologize for not being able to perform the last revision properly. Thank you again for your review and professional suggestions and comments.

Reviewer 4 Report

I am not sure if the authors have made the changes to the manuscript based on the 3 recieved reviews, or they have decided to withdraw their paper from the journal Molecules?

The changes to the manuscript are only minor.

The introduction has not been shortened, the results of the SPE have not been added, the results of the real samples have not been added.

The corrections have not been made (i.e. Figure 3 - Pointing out, again, this is not total ion chromatogram (a), these are overlayed extracted ion chromatograms. Even your picture calls it XIC=EIC, the same as in b-i)

For some unexplained reason, the Recovery values in table 4 are now different , as well as in Table 5.

I do not recommend this paper to be published in Molecules before all reviewers comments are met. 

Author Response

I am not sure if the authors have made the changes to the manuscript based on the 3 recieved reviews, or they have decided to withdraw their paper from the journal Molecules?

The changes to the manuscript are only minor.

Answer: We apologize for not resolving all the issues raised with the last revision, and we thank the reviewer for another review and for the professional suggestions and comments. We hope that this version will meet the reviewer’s requirements.

The introduction has not been shortened, the results of the SPE have not been added, the results of the real samples have not been added.

Answer: The content of the Introduction section has been streamlined as suggested by the reviewer, and all the changes are marked in red font.

The comparison of the recovery of the C18 and HLB SPE columns during the experiment was rough, and we did not perform a full experimental comparison of the recovery. We apologize for the lack of rigor. To avoid unscientific statements, we have changed the presentation of the relevant content; please refer to lines 175-176. When the filler loses the solvent environment, C18 tends to curl up and loses some extractive activity, which means that it has poor desiccation resistance. This disadvantage means that the experimental operation is more demanding in the SPE process. We hope this discussion will not cause any doubt or dissatisfaction.

The results for the real samples are described in lines 299-300.

The corrections have not been made (i.e. Figure 3 - Pointing out, again, this is not total ion chromatogram (a), these are overlayed extracted ion chromatograms. Even your picture calls it XIC=EIC, the same as in b-i)

Answer: We have changed “total ion chromatogram” in the heading of Figure 3 to “overlayed extracted ion chromatogram” as suggested by the reviewer.

For some unexplained reason, the Recovery values in table 4 are now different, as well as in Table 5.

Answer: The changes in the data shown in Tables 4 and 5 are based on suggestions from other reviewers, and we apologize for the confusion caused by our lack of clarity.

I do not recommend this paper to be published in Molecules before all reviewers comments are met.

Answer: We have thoroughly reviewed the manuscript and revised the contents with the hope that the new edition will meet the reviewer’s requirements. Thank you again for your professional and quick review.